# Older Barbary macaques show limited capacity for self-regulation to avoid hazardous social interactions

Eva-Maria Rathke[1,2,3], Roger Mundry[1,2,3] & Julia Fischer [1,2,3✉]

According to the Strength-and-Vulnerability-Integration (SAVI) model, older people are more motivated to avoid negative affect and high arousal than younger people. To explore the biological roots of this effect, we investigate communicative interactions and social information processing in Barbary macaques (*Macaca sylvanus*) living at 'La Forêt des Singes' in Rocamadour, France. The study combines an analysis of the production of ($N = 8185$ signals, 84 signallers) and responses to communicative signals ($N = 3672$ events, 84 receivers) with a field experiment ($N = 166$ trials, 45 subjects). Here we show that older monkeys are not more likely to specifically ignore negative social information or to employ avoidance strategies in stressful situations, although they are overall less sociable. We suggest that the monkeys have only a limited capacity for self-regulation within social interactions and rather rely on general avoidance strategies to decrease the risk of potentially hazardous social interactions.

[1] Cognitive Ethology Laboratory, German Primate Center, Leibniz Institute for Primate Research, Kellnerweg 4, 37077 Göttingen, Germany. [2] Department for Primate Cognition, Georg-August-University Göttingen, Göttingen, Germany. [3] Leibniz ScienceCampus Primate Cognition, Göttingen, Germany.
✉email: jfischer@dpz.eu

Older humans report higher life satisfaction than persons in mid-adulthood e.g. ref. [1]. A number of prominent life-span developmental theories have aimed to explain why this is the case. Social selectivity theory (SST) focuses on the importance of a limited future time perspective. According to SST, a shrinking time horizon affects the goals that people set for themselves, and favours behaviours that enhance well-being[2]. For instance, in the face of limited future time, people tend to focus on meaningful social partners and satisfying activities[2]. The Strength-and-Vulnerability-Integration (SAVI) model empha- sises that older humans aim to avoid situations that could potentially lead to adverse outcomes associated with higher arousal[3]. The SAVI model suggests that older humans avoid interpersonal conflicts and potentially stressful situations by shifting their attention away from the stressor or waiting for the situation to be resolved[4]. Both SST and the SAVI model are empirically well supported[5,6]. The avoidance of negative experi- ences is considered a result of self-regulation processes[7]. Both bodies of theory imply a sophisticated conception of time and may also involve meta-cognitive skills. Shifting preferences might also be related to age-related changes in the internal reward system[8], however, raising the question of to which degree bio- logical ageing (senescence) contributes to the observed age- related shifts in human preferences.

Studying ageing nonhuman primates (hereafter 'primates') allows distinguishing the importance of higher-level cognitive insight from more basic biological processes that contribute to motivational changes during the life span. Primates undergo similar physiological changes during aging as humans[9,10], but there is no evidence that they are aware of their limited future time[11]. Moreover, their social behaviour is not affected by cultural norms[12]. Studies of age-related changes in primate social beha- viour thus provide the opportunity to put some of the assump- tions of life-span psychological theories to a test[13–15].

Here, we set out to test hypotheses derived from life-span psychological theories that focus on self-regulation in a nonhu- man primate species, the Barbary macaques (*Macaca sylvanus*). With increasing age, female Barbary macaques have fewer part- ners. They engage in fewer but more extended affiliative inter- actions, which has been taken as evidence of an age-related increase in social selectivity[13,16]. Males experience changes in sociality similar to those experienced by females[17]. But how precisely do older individuals manoeuvre in their social groups? Is there evidence that older individuals strategically avoid negative affect, similar to humans? Building on previous studies, specifi- cally the investigation of age-related variation in the occurrence of affiliative and agonistic social interactions in females[16], and age- related variation in male and female social network position[17], we here focussed on the production and responses to communicative signals in both males and females. Communicative signals typi- cally function to initiate or deter subsequent physical interactions such as grooming or physical fighting. Analysing both the usage of signals and the contingency between signal and outcome, that is, whether and in which way the receiver responds to a given signal, may provide nuanced insights into age-related changes in the motivation to initiate or avoid physical interactions. To address our research question, we combined analysis of natural signal exchanges with a field experiment in which we investigated age-related variation in the interest in negative social information. Note that we distinguish between the motivation to acquire social information ('social interest') and the motivation to engage in physical social interactions. We assume that social interest is a precondition for the willingness to engage in physical social interaction, but is not necessarily tied to it. In other words, the animals may show social interest but may not be motivated to come into body contact, groom or fight with others. We assume

that the exchange of social signals plays a decisive role in increasing or decreasing the likelihood of physical interactions.

There are two ways in which communicative behaviour may contribute to altered social behaviour: signallers may change the propensity with which they produce specific signals, and receivers may differ in the propensity and types of responses to particular signals. We expected that older Barbary macaques would be less likely to initiate social interactions using communicative signals than younger monkeys. Following the SAVI model, we predicted that this effect would be more pronounced for agonistic signals. We further predicted that older monkeys would be less responsive to other monkeys' signals. Again, following the SAVI model, we expected that the effect would be more pronounced for agonistic signals. Specifically, we expected that older monkeys would be more likely to adopt avoidance or de-escalation strategies, such as ignoring others' signals or leaning or walking away (study 1). In addition, we conducted a field experiment in which we presented pictures of unknown conspecifics displaying agonistic ('open- mouth threat face') or neutral facial expressions (study 2). We predicted that older monkeys would spend less time looking at pictures depicting agonistic facial expressions. Although the SAVI model focusses on the avoidance of negative situations, we ana- lysed the production of and responses to both affiliative and agonistic signals, to be able to infer whether the monkeys speci- fically avoided negative interactions or social information, or whether they generally signalled or responded less.

Neither the analysis of the responses to signals nor of the responses in the experiment revealed evidence for a specific avoidance of negative compared to positive or neutral social signals. We suggest that the monkeys have only a limited capacity for self-regulation within social interactions.

## Results

The production of agonistic signals varied with age ($P < 0.001$) and sex ($P < 0.001$; model 1; likelihood ratio test for negative binomial models (full-null model comparison: LR statistic = 48.89, $df = 3$, $P < 0.001$; Fig. 1a and Table 1). More specifically, the production of agonistic signals was highest in mid-adulthood, and males pro- duced such signals more frequently than females. To illustrate the monkeys' behaviour, young males produced on average 2.43 ago- nistic signals/h, and young females produced 1.88 signals/h. In mid-adulthood, males produced 4.15 agonistic signals/h and females 2.65 signals/h. For old males, the rate of agonistic signals was 2.55 signals/h, and for old females, it was 1.25 signals/h. Effect sizes (Nagelkerke's $R^2$) were 0.13 for sex and 0.35 for the combined effects of age and age[2].

The production of affiliative signals also varied with age and sex (model 2; likelihood ratio test LR statistic = 33.85, $df = 2$, $P < 0.001$; Table 2). On average, young females produced affilia- tive signals more frequently than males, while mid and old males produced affiliative signals slightly more frequently than females (Fig. 1b). Across the sexes, young and mid-adult subjects pro- duced affiliative signals more frequently than old subjects. More specifically, young males produced on average 0.99 affiliative signals/h, and young females 2.39 signals/h. In mid-adulthood, males produced 1.43 affiliative signals/h, and females produced 1.21 signals/h. For old monkeys, the rate of affiliative signals was 0.96 signals/h for males and 0.55 signals/h for females. Effect sizes (Nagelkerke's $R^2$) were 0.06 for sex and 0.19 for age. Note that the estimate for males was negative (i.e. males should be less likely to produce affiliative signals than females). Yet, an inspection of the data shows that males were more likely to produce affiliative signals. This paradox can be explained by the fact that males were, on an average higher ranking than females, such that the positive effect of rank explained also the variation in the

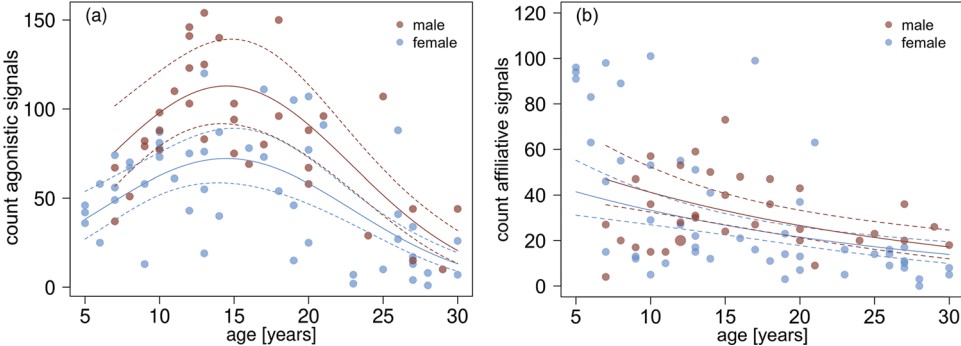

**Fig. 1 Variation in signal usage in relation to age. a** Total number of agonistic signals produced in relation to age (model 1, $N = 5485$ signals). **b** Total number of affiliative produced in relation to age (model 2, $N = 2700$ signals). Females ($N = 50$) are represented by blue, males ($N = 34$) by brown points. Point size represents the frequency of a signal at a given age (range 1 to 2). The solid lines depict the fitted model, and the dashed lines indicate their lower (2.5 %) and upper (97.5 %) confidence limits. The model shown in **b** is for an individual with an average rank (determined separately for females and males).

**Table 1 Influence of age and sex on the production of agonistic signals (model 1).**

| Term | Est. | SE | z value | CI (lower) | CI (upper) | P |
|---|---|---|---|---|---|---|
| (Intercept) | 1.01 | 0.11 | −0.11 | 0.80 | 1.22 | (a) |
| z.age [b] | −0.17 | 0.07 | −2.56 | −0.30 | −0.04 | <0.01 |
| z.age[b] | −0.38 | 0.07 | −5.39 | −0.51 | −0.24 | <0.001 |
| sex_male [c] | 0.45 | 0.13 | 3.58 | 0.20 | 0.70 | <0.001 |

Estimates (Est.) with standard error (SE), test statistic $z$, lower and upper confidence limit (CI), and P values are given ($N = 84$ subjects).
[a] not indicated because of very limited interpretability.
[b] $z$-transformed to a mean of zero and a standard deviation (sd) of one; mean and sd of the original variable were 16.1 and 7.2 years, respectively.
[c] dummy coded with the female being the reference category.

**Table 2 Influence of age, sex and rank on the production of affiliative signals (model 2).**

| Term | Est. | SE | z value | CI (lower) | CI (upper) | P |
|---|---|---|---|---|---|---|
| (Intercept) | −0.42 | 0.11 | −3.92 | −0.62 | −0.20 | (a) |
| z.age [b] | 0.31 | 0.07 | −4.19 | −0.45 | −0.17 | <0.001 |
| sex_male [c] | −0.49 | 0.21 | −2.37 | −0.89 | −0.09 | 0.02 |
| z.rank [d] | 0.44 | 0.11 | 4.08 | 0.24 | 0.64 | <0.001 |

Estimates (Est.) with standard error (SE), test statistic $z$, lower and upper confidence limits (CI) and P values are given.
[a] not indicated because of very limited interpretability ($N = 84$ subjects).
[b] $z$-transformed to a mean of zero and a standard deviation (sd) of one; mean and $sd$ of the original variable were 16.0 and 7.2 years, respectively.
[c] dummy coded with the female being the reference category.
[d] $z$-transformed to a mean of zero and a standard deviation (sd) of one; mean and sd of the original variable were 20.61 and 4.92, respectively.

likelihood to produce signals between males and females, with higher ranking animals being more likely to produce affiliative signals. In conjunction, males thus produced somewhat fewer signals than expected for their rank position.

In the following analysis, we investigated whether older animals were particularly likely to ignore agonistic signals. We found no evidence for the predicted interaction between receiver age and signal category (agonistic/affiliative) affecting the likelihood to respond (non-significant full-null model comparison: model 3, $\chi^2_2 = 1.83$, $P = 0.40$; Fig. 2). The probability of response was ca. 0.8 for affiliative as well as agonistic signals (Supplementary Table 4). Concerning the control fixed effects 'signaller age' and 'signaller sex', we found that the probability of responding clearly varied with signaller age; the older the signaller was, the less likely it was that the receiver responded to the signal (Fig. 3). The likelihood that females responded to a signal was 83% for the youngest monkeys' signals and 72% for middle-aged monkeys' signals. The response rate to old monkeys' signals was 57%. The likelihood of responding varied neither with signaller sex nor signal category (Supplementary Table 5).

The age of the subject (receiver) was not obviously related to the type of response shown, neither to affiliative nor agonistic signals. With regard to the control fixed effects 'signaller age' and 'signaller sex', we found that subjects were more likely to respond with 'Teeth-chatter', a low-cost submissive signal, in response to agonistic signals by females (42% of $N = 860$ cases) than to agonistic signals by males (28% of cases). In turn, subjects were more likely to lean or walk away following male agonistic signals (combined 'give ground' and 'make room': 70% of $N = 850$ cases) compared to female agonistic signals (55% of cases). In the

multinomial model, we also tested whether the different response types varied differentially with age. Old females were more likely to respond to an agonistic signal with 'Teeth-chatter' compared to younger females, while there was no age effect for the other signal categories (Supplementary Fig. 1).

In the field experiment, the median duration for the initial look for agonistic pictures was 1.32 s (range: 0.12–16.4 s); for neutral pictures, it was 1.16 (range: 0.04–10.9 s); the median total looking time duration for agonistic pictures was 2.64 s (range 0.4–43.4 s) and 2.24 s (0.04–33.0 s) for neutral pictures. Neither the interaction between age and picture type nor age or picture type were clearly related to variation in initial looking time (Fig. 4) or total looking time. We found no significant differences between the full and null models (initial looking time: $\chi^2_2 = 1.56$, $P = 0.457$; total looking time $\chi^2_2 = 1.02$, $P = 0.60$). In the analysis of total looking time, the number of presentations (first or second pair) had a significant effect on looking time, with looking time being significantly shorter for the second pair; see Supplementary Tables 6–9 for the model outputs for all analyses of looking time). Younger individuals were more likely to touch or sniff at the picture presented (Supplementary Fig. 2).

## Discussion

Older monkeys produced fewer affiliative signals, such as teeth chattering and lip-smacking, than younger monkeys. The variation in signal production corresponds to the variation in affiliative signals involving physical interactions[16,17]. Young females showed the highest rates of affiliative signals, suggesting that they have the highest motivation to establish and consolidate social

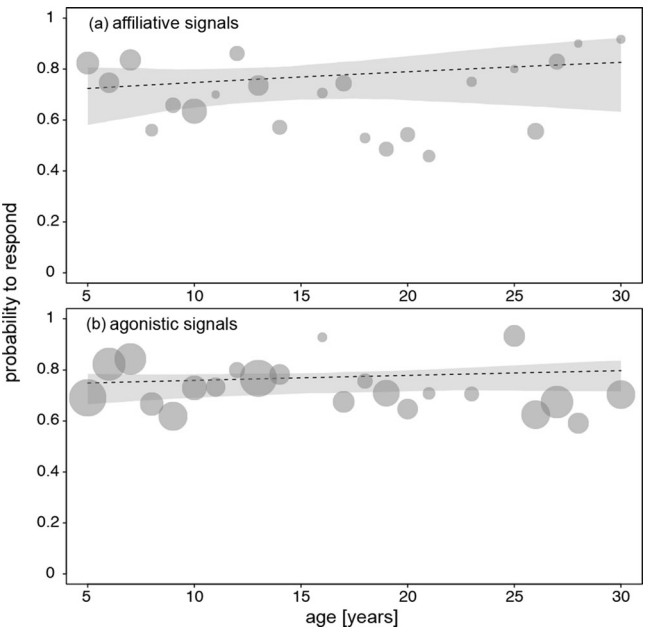

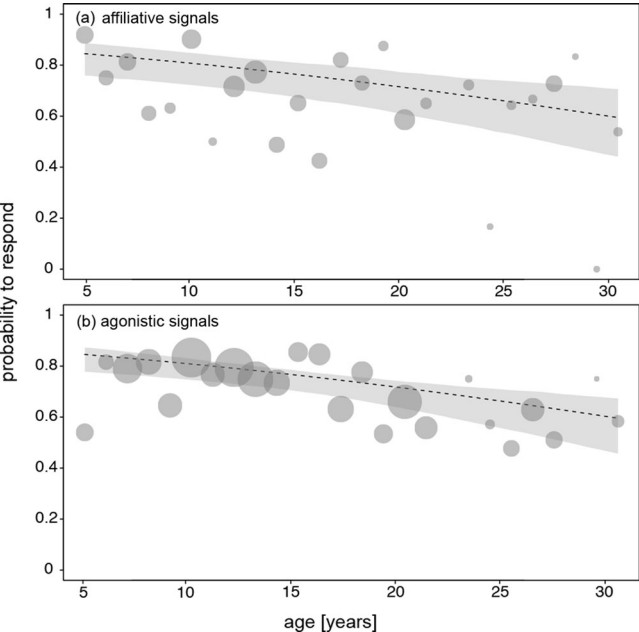

**Fig. 2 Effect of receiver age on the probability of females to respond to a signal. a** Probability to respond to affiliative signals in relation to receiver age ($N = 846$ events involving 50 female receivers). **b** Probability to respond to agonistic signals in relation to receiver age ($N = 2269$ events involving 50 female receivers). The area of the points represents the number of events per age (range: 10 to 233). Dashed lines and grey polygons indicate the fitted model and its (2.5 and 97.5%) confidence limits (for all other terms in the model being centred to a mean of zero).

**Fig. 3 Effect of signaller age on the probability of females to respond to a signal. a** Probability to respond to affiliative signals in relation to signaller age ($N = 846$ events involving 50 female receivers). **b** Probability to respond to agonistic signals in relation to signaller age ($N = 2269$ events involving 50 female receivers). The area of the points represents the number of events per age (range: 4 to 264). Dashed lines and the grey polygons indicate the fitted model and its (2.5 and 97.5%) confidence limits (for all other terms in the model being centred to a mean of zero).

bonds. In line with previous studies, males produced agonistic signals more frequently than females. The age-related trajectory in the production of agonistic signals by males matches the variation in resource holding potential, with a peak at approximately 15 years of age[18,19]. Similarly, middle-aged females are typically higher ranking than young or old females, and the production of agonistic signals corresponds to their rank position.

Concerning the responses to other group members' signals, our results did not conform to the predictions of the SAVI model. We did not find the predicted interaction between age and signal type in the responses to group members' signals, as older monkeys were not more likely to ignore or move away from agonistic signals as a strategy to regulate negative affect or avoid potentially costly interactions.

In the field experiment, we did not find the predicted interaction between age and facial expression category either, suggesting that older monkeys did not specifically avoid negative social information. We are relatively confident that the lack of distinct responses is not due to issues with the methodology, as this method had been used to reveal differential interest in out-group vs. in-group conspecifics[20] and babies and friends vs. non-friends[13] in this population. Yet, it could be possible that the animals were not able to distinguish different facial expressions when shown in a picture. From our data alone, we are unable to decide whether the animals simply did not or were indeed unable to distinguish between the different facial expressions. The pattern observed in the Barbary macaques differed from that reported for rhesus monkeys, *Macaca mulatta*. In this species, a comparable study involving the presentation of photographs showing male and female monkeys with different facial expressions, older monkeys looked less at the pictures than younger monkeys, but the age-related decrease was attenuated for the threat photo[14]. Importantly, these results did not conform to the predictions of the SAVI model either. Given

that rhesus monkeys responded differentially to different facial expressions depicted in photos[14], we assume that the lack of a distinction in Barbary macaques is not due to the fact that they would not be able to distinguish between pictures showing neutral and agonistic faces.

Rhesus macaques differ from Barbary macaques in terms of their social tolerance and are classified as a rather despotic species[21]. Interestingly, female rhesus macaques showed selective attention to agonistic interactions of third parties compared to affiliative interactions[22]. Barbary macaques, in contrast, live in a relatively egalitarian system[23] with a significant share of ambivalent relationships and ambiguous social signals[24]. Coalitionary support by others is a major determinant of conflict outcomes[19,25]. Therefore, agonistic facial expressions may elicit less attention in Barbary macaques than in rhesus macaques. Our study lends further support to the notion that differences in social structure, including the quality of relationships and steepness of the rank hierarchy, shape the allocation of social attention[26]. In addition, the lack of an age effect in the field experiment corroborates previous findings in this study population[13] that the interest in social information remains stable, although the rate of social interactions declines. The results underscore the importance of distinguishing between the motivation to engage in potentially detrimental social interactions and the motivation to obtain social information.

Taken together, the social reclusion of older males and females appears to result from two processes, driven by younger individuals on the one hand and the old individuals on the other. First, older monkeys are less often the targets of interactions and interact with fewer partners[17]. Second, signals produced by older monkeys were more likely to be ignored by other group members, suggesting that older monkeys are perceived both as less threatening and less valuable as social partners. However, old

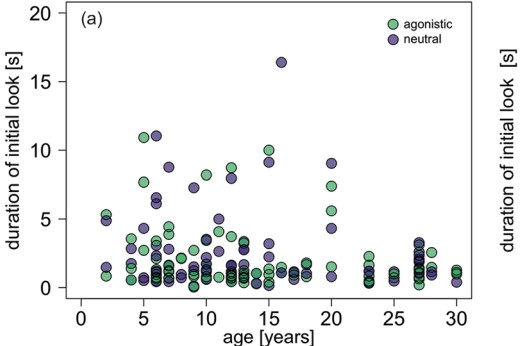
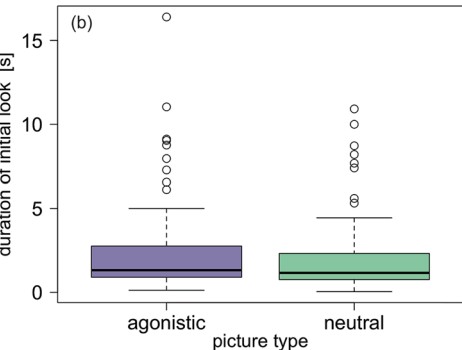

**Fig. 4 Time spent looking at agonistic and neutral pictures in relation to age and condition. a** Initial looking time in relation to age and picture type (violet = agonistic; green = neutral). **b** Initial looking time in relation to picture type (agonistic, neutral). Boxplots with median and interquartile range based on 166 trials with 45 subjects. Whiskers show values within 1.5 times the interquartile range. Dots indicate individual values.

individuals may maintain specific relationships with selected partners[13]. Detailed analyses of the long-term development of dyadic relationships will be needed to explore the differentiated production of and responses to signals with regard to the relationship quality of a dyad.

Older monkeys did not specifically ignore negative signals in the behavioural observations and they also did not specifically avoid social information in the field experiment. Thus, the predictions from the SAVI model were not met. In the behavioural observations, we also found no evidence for a 'positivity effect'[27], according to which older individuals favour positive information and avoid negative information. Rather than using signals strategically with the aim of shaping the kind of interaction, older monkeys appeared to avoid physical interactions more generally[16,17].

One could raise the question to which degree the monkeys' signalling and response patterns were influenced by the living conditions. It may well be that the overall interaction frequencies are lower in the wild compared to captivity. In the wild, the animals need to spend more time foraging or in group movement. Yet, it seems unlikely that the contingencies between signals are responses would differ substantially, as they appear to be rather fixed across age-classes. Note, however, that an equivalent analysis would not be possible in the wild, as extremely old animals are hardly found under natural conditions.

Overall, our results, as well as those by Rosati and colleagues[15], suggest that the general motivation to engage in social interactions declines with age in both humans and different nonhuman primate species, although the motivation to acquire social information does not vary with age (this and other studies on nonhuman primates). With regard to the management of social interactions, the available evidence suggests that only humans are able to employ more sophisticated self-regulation strategies in old age. For instance, the model of Selection, Optimisation, and Compensation[28] stresses the importance of active goal setting, the use of adaptive strategies to attain these goals, and the switching to alternative strategies when previous strategies are no longer efficient[29,30]. Likewise, older adults are assumed to employ cognitive control mechanisms to regulate their emotions[27]. It appears unlikely that nonhuman primates have such control mechanisms at their disposal; instead, they very much "live in the moment", and age-related changes in sociality or problem solving are related to changes in motivation[13].

Interestingly, a recent large-scale survey involving 1.7 million respondents in 166 countries observed only small age-related differences in negative affect or life satisfaction in humans across the life span but a substantial decrease in positive affect with increasing age[31]. That is, a central tenet in life span developmental theories has been called into question. At the same time, substantial differences between different cultural regions were found. The differences between different cultures – but also between humans and nonhuman primates – highlight the need for further research on the question of how affective experience, emotion regulation, and emotion perception vary with age, individual experience, and cultural background[32]. Future studies should involve experimental paradigms that can be applied to both humans and nonhuman primates to develop a comprehensive understanding of how evolved biological processes and cognitive evaluation interact and contribute to emotion regulation.

## Methods
We conducted this study in 2017 and 2018 in the enclosure 'La Forêt des Singes' in Rocamadour, France[33]. During the study period, 170–180 Barbary macaques aged between 0–30 years (see Supplementary Table 1) lived in three social groups in the park. Data collection took place from April to June and from September to November in two of the three groups 5 to 6 days a week, from 9 a.m. to 8 p.m. Animals in the PB group were observed in 2017; animals in the GB group were observed in 2018. We considered monkeys 'young' up to an age of 9 years, 'middle-aged' between 10 and 19 years and 'old' when they were 20 years and older. Note that in the statistical analyses, age was always entered as a continuous variable.

**Study 1.** Together with a total of four field assistants (two per season), EMR collected behavioural observations from all females ≥5 years old and all males ≥7 years old (total $N = 84$ subjects; $N = 50$ females) in two of the three social groups living in the park. The analyses are based on an average of 26.0 observation hours for each subject (range: 24 h 38'-27 h 25', except for one male who died and was observed for 12 h 31', resulting in a total of 2180 observation hours. The focal observations were evenly balanced across subjects, season, day times and observers. The mean observer reliability during 21 simultaneous focal observations involving all observers and both seasons was 0.86 (intraclass correlation coefficient (ICC(1,k) from the R package *irr*). We collected 30-min continuous focal protocols[34] using handheld devices (Samsung Galaxy Note 2) with the software programme Pendragon Forms (Pendragon Software Cooperation, Libertyville, IL, USA). During focal observations, we recorded all social interactions of the focal animal and extracted all instances of the production of agonistic and affiliative signals. Agonistic signals included threat stare, open-mouth threat, head bob, ground slap, silent scream face, and scream face; affiliative signals included teeth-chatter and lip-smack. In addition, we noted the responses of the interaction partner. We further recorded additional agonistic interactions ad libitum to establish the dominance hierarchies. To this end, we used all dyadic and decided agonistic interactions (submissive reaction and no counter-aggression). We determined the dominance rank based on the normalised David's score, implemented in the *EloRating* package in R;[35] for further details, see ref. [17].

**Study 2.** We conducted a field experiment in the GB group from April to June 2018. We tested 47 monkeys, including 25 adult females (5 to 30 years old), 13 adult males (7–27 years old) and nine juvenile/sub-adult males (2–6 years old). Following the procedures described in[20], we presented the monkeys with photographs of conspecifics with different types of facial expressions and measured the time spent looking at the picture. Pictures were taken from members of the other groups in the park. As individuals of the social groups rarely interact with one another, we did not need to control for the animals' social relationships in the analysis. We took 26 pairs of pictures (11 from females, 15 from males) of neutral

**Fig. 5 Examples of stimuli used in the experiment. a** Example of a female agonistic and neutral facial expression. **b** Example of a male agonistic and neutral facial expression. Note that in contrast to other macaque species such as rhesus monkeys, Barbary macaques do not produce 'coos'. Thus, the pictures shown can be unequivocally categorised as open-mouth threat faces. Photographs were taken by Eva-Maria Rathke.

and mildly agonistic facial expressions ('open-mouth threat face') using a Nikon E-300 photo camera. We printed the photos on matte DIN-A4 paper with a 0.17 m diameter and used each picture two to five times. In total, we presented every subject with up to two pairs of photos (one male and one female pair). The identity of the subject on the photo was constant within a pair (Fig. 5). Within each pair, we balanced the order of presentation for the type of facial expression (agonistic/neutral) and randomised the assignment of the picture pairs to the subjects.

For any given trial, the photo was placed in a wooden frame with two plastic rails to keep it in place. The experimenter was blind to the type of facial expression. The experimenter sat down approximately three metres away from the test subject with the wooden frame and a photo inside. White cardboard covered the photo before the test started. During the testing procedure, the experimenter wore a baseball cap and sunglasses to avoid eye contact with the subject. The experimenter began filming the scene using a Panasonic HC-X929 video camera and attracted the subject's attention by tapping against the wooden frame. Once the subject looked up or in the direction of the photo, the experimenter removed the cover and filmed the monkey's response for one minute. When the monkey left the test area (1-m radius around the test setting), the trial ended. After each trial, there had to be at least a break of four days before testing the same subject to avoid habituation to the testing paradigm.

We conducted a total of $N = 177$ trials with $N = 47$ subjects. Eleven trials had to be excluded from the analysis, either because the pairwise presentation could not be completed ($N = 9$ trials) or because of experimenter error ($N = 2$ trials), resulting in 166 trials with 45 subjects for analysis. Supplementary movie 1 provides an example of a subject's response.

To analyse the responses in the field experiment, we assessed looking time by examining the videos frame-by-frame with 25 frames per second with the programme Mangold Interact (Version 17). We scored the duration of the 'first look', i.e. the time from looking at the photo until the animal looked away for the first time, to measure initial interest, and scored the total looking time within the first minute as a measure of the overall interest. Additionally, we recorded the occurrence of self-directed behaviours (yawn, scratch, or self-grooming), communicative signals (lifting eye-brows, head bob, lip-smack) and picture manipulation (touch) but used these only for descriptive purposes. The interobserver reliability was assessed for $N = 56$ of the video clips using the intraclass reliability correlation coefficient (ICC(1,k) from the R package *irr*. The agreement was excellent[36] for looking time (0.98). For the behaviour towards the pictures, the observers agreed in all cases.

**Statistics and reproducibility**. For the analysis of signal production, we applied a general linear model (GLM) with a negative binomial error distribution and logit link function, with the function *glm.nb* of the R package *MASS*. We conducted all analyses in the R environment (see Supplementary Table 2 for all version numbers). A Poisson distribution did not provide a good fit, as both response variables appeared overdispersed given the model. Model 1 comprised the analysis of the production of agonistic signals; model 2 comprised the analysis of affiliative signals (total number of signals per signaller in both models). Age and rank were z-transformed to a mean of zero and a standard deviation of one to ease the interpretability of the model estimates. Rank was only included in the analysis of affiliative signals but not used in the analysis of agonistic signals because the dominance rank was based on the occurrence of agonistic signals and further agonistic behaviours such as lunge, chase and physical fighting. Hence, the inclusion of rank would be entirely circular. We included focal observation time (log-transformed) as an offset term[37]. Although the response was a count in both models, by including focal time as an offset term, we effectively modelled signalling rate (i.e. number of signals per unit time). We checked the stability of both models using the function *dfbeta*, dropping individuals one by one, and assessed potential collinearity issues by determining variance inflation factors (VIFs) using the function *vif*[38] of the R package *car*.

The analysis of the production of agonistic signals revealed that a quadratic relationship better predicted age-related variation in the number of signals. However, it should be kept in mind that the inclusion of age squared represents an a posteriori hypothesis. Hence, caution is required when interpreting such adjusted

models. To avoid 'cryptic multiple testing'[39] and keep the type I error rate at the nominal level of 0.05, we compared both full models with a respective null model which lacked age (for both models), age$^2$ and rank (for model 1).

The sample analysed for these models comprised the production of 5485 agonistic signals (model 1) and 2700 affiliative signals (model 2), recorded from 84 signallers (50 female). Neither of the two models was overdispersed (dispersion parameters, model 1: 0.942; model 2: 1.032), and collinearity was also not an issue (maximum VIF, model 1: 1.001; model 2: 2.692). Both models were also of good (model 1) or moderate stability (model 2) as assessed by means of DFBeta values. We determined Nagelkerke's $R^2$ as a measure of effect size[38] by comparing the log-likelihood of the full model with those of reduced models lacking the predictor variable in question.

With model 3, we estimated the extent to which the probability of showing any response (yes/no) was influenced by receiver age. This model was run for females only, as we only observed 557 events involving male receivers, compared to 3115 events involving 50 female receivers. We fitted a generalised linear mixed model (GLMM)[40] with binomial error structure and logit link function[37]. We included receiver age and its interaction with the signal category (agonistic or affiliative) as our key test predictors with fixed effects. We also included the age and sex of the signaller and the main effect of signal category as control fixed effects.

We included random intercept effects for the identity of the receiver, the signaller, and the receiver-signaller dyad to avoid pseudoreplication. It has repeatedly been shown that omitting random slope effects that could potentially affect the response leads to a greatly inflated type I error rate in inference about the fixed effects[41–43]. To prevent such overconfident estimates and keep the type I error rate at the nominal level of 0.05, we included all theoretically identifiable random slopes[42,43], namely those of receiver age, signal category, and their interaction within the signaller and those of signal category, signaller age, and signaller sex within the receiver. In terms of their biological meaning, these random slopes estimate the extent to which the effect of a predictor varies between signallers or receivers. For instance, the random slope of receiver age within the signaller takes into consideration the possibility that the responsiveness to a signal will vary in an age-related fashion with the identity of the signaller (e.g. older subjects will be more or less likely to respond to subject A vs. subject B). Similarly, the random slope of signaller age within the receiver estimates how much a potential effect of signaller age on responsiveness varies between receivers. Importantly, not including the respective random slope means to make the strong (unlikely) assumption that, for instance, the effect of signaller age on responsiveness is exactly the same for all receivers. We also included parameters for the correlations among random intercepts and slopes. To avoid cryptic multiple testing[39], we compared this full model with a null model that lacked receiver age and its interaction with signal category in the fixed-effects part.

The model was fitted using the function *glmer* of the R package *lme4*. Prior to fitting the model, we z-transformed receiver age and signaller age to achieve a more straightforward interpretation of the estimates and to ease model convergence. We manually dummy coded and then centred signal category and signaller sex before including them as random slopes. We determined confidence intervals of model estimates and fitted values by means of a parametric bootstrap (function *bootMer* of the R package *lme4*; 1000 bootstraps). Significance of individual effects we obtained by dropping them from the full model, one at a time, and comparing the respective reduced models with the full model. All model comparisons were based on likelihood ratio tests[44]. To estimate model stability, we excluded individual signallers, receivers, and dyads one at a time from the data set, fitted the full model to each of the subsets and compared the estimates derived with those for the full data set. The model had good stability in the fixed-effects part and did not suffer from collinearity[38], as indicated by a maximum variance inflation factor of 1.023 (based on a model lacking the interaction).

The sample analysed for this model comprised a total of 3115 events where we noted the responses of females ($N = 846$ to affiliative and $N = 2269$ to agonistic signals). Signals were given by 83 signallers to 50 receivers, which together formed 1005 signaller-receiver dyads. We observed a total of $N = 2238$ behavioural reactions and $N = 877$ 'no response'. For male receivers, we observed a total of $N = 557$ events (279 affiliative and 278 agonistic signals). Males showed no responses to other group members' signals in 85 affiliative signalling events and 158 agonistic signalling events. Due to the smaller sample size and the model complexity, we refrained from further analyses of male receiver behaviour.

With model 4, we addressed which types of response individuals produced after a group member's agonistic signal and how this choice was affected by receiver age. As above, we included the signaller's age and sex as control factors. As response types, we included the patterns 'Give Ground', 'Make Room', 'Present', and 'Teeth-Chatter', as these occurred with sufficient frequency (>25, Supplementary Table 3). The fitted model was identical to the model of 'any response' (model 3; with the exception that it lacked 'signal category' in the fixed as well as random-effects part). However, the response in this model was a categorical variable with four states. Hence, we fitted a multinomial model, which can be conceived as a generalisation of the logistic model suited for a response comprising more than two states[45]. Since the response was multinomial and since we were not aware of an option to fit such a model with complex random effects structure in a maximum likelihood framework, we decided to use a Bayesian framework and applied the function *brm* of the R package *brms*. We fitted the model with a maximum tree depth of 20 and set the adapt delta to 0.99. The chains successfully converged, as indicated by Rhat

values between 1.000 and 1.001. The sample considered for this model comprised a total of 1594 responses by 50 receivers in response to signals of 81 signallers; signallers and receivers formed 654 dyads. We did not conduct a separate analysis for response types after affiliative signals, as the two types that occurred most frequently were relatively similar facial expressions ('Lip-Smack' and 'Teeth-Chatter').

To estimate the effects of age on the looking time in the field experiment, we fitted linear mixed models [LMM; 40] using the function *lmer* from the package *lme4*[46]. We fitted one model for the initial looking time (model 5a) and one for the total looking time (model 5b). We included subject age, subject sex, picture type (agonistic/neutral), the interaction between subject age and picture type, and sex of the subject shown in the picture as fixed effects and the IDs of the test subject and of the subject shown in the picture as random effects. We included random slopes of picture sex and picture type within the test subject ID and age, picture type, and the interaction between age and picture type within the picture ID. We used a likelihood ratio test to compare the full model, including all predictors, with the null model lacking the predictors of interest.

We checked whether the assumptions of normally distributed and homogeneous residuals were fulfilled by visual inspection of a qq-plot of the residuals and residuals plotted against fitted values. The inspection of the correlations between the predicted values and the residuals revealed that it was not ideal (slight positive correlations), most likely due to the small number of repeated measures per individual. Given the absence of any strong effects apparent in the data, we did not believe that the deviation affects our conclusions. As above, we determined variance inflation factors using the R package *car*. None of the VIFs exceeded 2.0, thus raising no concerns.

The study complies with the Guidelines for the treatment of animals in behavioural research and teaching (Animal Behaviour 2020, Volume 159, I-XI) and the rules and regulations of the countries in which the research was conducted. Due to the observational nature and the setting of the study, no specific ethical approval was obtained prior to the beginning of the study. The government of Lower Saxony had confirmed that such studies do not require approval according to the Animal Care Act (Document No. 33.19-42502-04 from 28.09.2016).

**Reporting summary**. Further information on research design is available in the Nature Research Reporting Summary linked to this article.

## Data availability
Data and source files for Figs. 1–4 are available at[47] https://doi.org/10.17605/OSF.IO/VJEB3.

## Code availability
R scripts and functions for all statistical analyses are available at[47] (https://doi.org/10.17605/OSF.IO/VJEB3).

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

## Acknowledgements
The study was funded by the Deutsche Forschungsgemeinschaft [DFG, German Research Foundation, project number 360742713 (Fi707/22-1) and 254142454/GRK 2070] and a Seed fund by the Leibniz ScienceCampus Primate Cognition, funded by the Leibniz Association. We thank Ellen Merz, Gilbert de Turckheim, and Guillaume de Turckheim for the permission to conduct this study at La Forêt des Singes and the park staff for their support. We are grateful to Magdalena Wimmer, Luisa Beckmann, Lateefah Roth and Marie-Laure Poiret for their help in collecting the data. Nana Hesler provided inspiration for this study and Alexandra Freund helpful comments on the manuscript.

## Author contributions
E.-M.R. developed the study concept and design, collected the data, performed statistical analyses, prepared the figures and drafted the manuscript. R.M. performed statistical analyses and drafted the manuscript. J.F. developed the study concept and design, performed the statistical analysis, prepared the figures, drafted the manuscript and compiled the final version.

## Funding

## Competing interests
The authors declare no competing interests.
