## [Peer Review File · Communications Biology]

Reviewers' comments:

Reviewer #1 (Remarks to the Author):

This is a thoughtful and detailed study examining age-related changes in sociality in macaques. The study appears to be well-powered to support either no or a very tiny effect of age. Overall the manuscript is well-written and detailed (sometimes too detailed). The major comments relate to the justification and interpretation of this study in the context of two similar previous studies from the same group and the impact of the modeling decisions the authors made, which are detailed below.

Overall comments:

This work builds upon previous studies from this group showing decreased motivation, but not interest, for social interactions with age (<https://pubmed.ncbi.nlm.nih.gov/28984992/>, <https://pubmed.ncbi.nlm.nih.gov/27345168/>). The novel component here seems to be the focus on the avoidance of agonistic signals, not just interactions, but that is not clearly justified in the introduction. Overall, it is unclear why the authors expect a decrease in initiation of all interactions, including affiliative, given their previous findings and, importantly, how this fits with the SAVI model they are testing. It might be better to focus just on the negative interactions, which are the crux of their test of the SAVI model. Alternatively, if the authors wish to continue to present changes in affiliative signals then better justification for why both production of and responsiveness to affiliative signals are predicted to decline with age based on SST theory and/or the author's previous findings is needed. Additionally, it is important that the authors clearly state how this study is different from their previous study(ies) – especially their 2017 paper. In the discussion the authors should clearly explain how these findings either support, extend, or counter their previous findings.

There is a lot of detail about the statistical approach, almost too much, which makes the text a bit more confusing than necessary. For instance, in L139-142 (and elsewhere), there is discussion of a number of different random effects and slopes nested within them – this is a lot of parameters in one model. More importantly, it is unclear why all of these random effects and slopes were included and why none of the slopes themselves were discussed in the results (which is important for interpretation). These random slopes were included in some of the null models, which might explain the lack of a main/fixed effect in the full models. On a related note: what is an “overconfident model”? Why is the signaler age a random slope within the receiver and vice versa? It is even unclear from the code, which is great and available on OSF. Overall these statistical decisions should be justified and clarified – especially when the authors want to report null findings.

Minor: comments:

What is the difference between motivation and interest? And how does that play into this study and their prior work?

L108: “Signal usage” is a bit of a vague title, the focus here is on the production of affiliative/agonistic signals, so perhaps something like “Signal production” might be more informative for the reader.

L132: Similar to the comment above, a more informative title would be useful here, something like “Probability of responding to affiliative and agonistic signals”. Along the same lines, whatever title is chosen here should align with the title given for that section in the Results (i.e. L208) so that the reader can easily follow the moving pieces of this study.

L152-156: Did you remove all observations from one individual (or dyad) and then rerun the model to assess stability or did you remove the covariates themselves and assess stability? It is unclear.

L192: “Varied” in what way? There is no indication that there is a quadratic pattern in how the

production of agonistic signals changes with age; the reader has to look to Figure 1 to get this information. On the whole, the figures were clear, but the text was difficult to parse and understand. Better descriptions of the results are needed here and elsewhere in text (e.g. L199). Along the same lines, it would be preferable to present the effect sizes and the p-values for those effect sizes, not just the LRTs for model comparisons. Overall, the results need to focus on the effects, not the model comparisons.

L209: The wording here seems misleading. It sounds like there is no overall effect of age (when there is) when what is really meant is that there was no significant interaction between age and signal category. Maybe simplify to what is stated on L210.

L218 -227: This section is generally confusing and requires some additional clarification. Why is Fig. 2 referenced here? This figure presents the results from the analysis looking at the probability of responding, not the type of response. Similarly, where did "teeth chatter" come from? In the methods the four types of responses that the authors indicate they are looking at following an agonistic signal are "Give Ground, Make Room, Present, and Lip Smack". On L224-227 the authors switch back to discussing the "likelihood of responding" which, again, is not what these analyses are about.

Figure 2 & 3: Minor comment: put labels on the actual figure panels. E.g., "a) affiliative." so that the reader doesn't have to go through the legend to completely understand

Figure 4. To someone who studies rhesus macaques, those look like affiliative images (e.g., "coos"), and not agonistic ones (e.g., open-mouth threats). Especially the female example photo. If both these agonistic photos are indeed open-mouth threats then perhaps a bit more context is warranted for what these behaviors look like in this species?

L299-301: Could the authors provide some additional descriptive stats for duration of first look and total looking time (e.g. min,max,mean)?

L313: Why is "picture type" included as a random effect nested within signaller? And what is the difference between Picture ID and Signaller ID in this context? Should one of these be Subject ID? It is not clear.

L323: Should this read, "thus raising no concerns"?

L324: This is repetitive as the same thing is stated on L315-316.

L328: The wording here is again confusing. This is another instance where it would be helpful to clearly state each result and provide the appropriate parameter estimates and p-values. Also, picture type is a random slope within individuals, but then also a fixed effect. Why?

Fig. 5. Y-axis label should be "duration of first look" not "looking time"

L350: Is there not an alternative way of looking at these results? If the idea is to avoid negative/agonistic interactions to avoid high arousal and the associated negative outcomes then surely you actually need to attend to those cues more strongly (or at least not attend to them less)? So might not continued attentiveness to agonistic signals with age be an indication that older individuals are actively trying to avoid negative interactions?

Reviewer #2 (Remarks to the Author):

This is a clearly written and well organized paper. The results have the potential to make a useful contribution to social aging in primates and provide important evolutionary context for patterns of human aging.

I have two major comments that I would like to see addressed in a revision. Below these, I list

several minor comments.

First main comment: In line 115, I was surprised that rank was not included in the model of agonistic signals. The goal of this analysis (model 1) was to test for age signatures in the use of affiliative and agonistic gestures. As the authors explain, rank will be an important predictor of agonism. However, rank is correlated with age in many species. For instance male rank and age are often correlated in polygynous primates such that young adult males have high rank, and rank falls in middle age and older adulthood. This rank pattern could explain the relationship between age and agonistic gestures seen in Fig. 1A. Therefore, it seems important to control for effects of rank in order accurately interpret the age effect on agonistic signals. At a minimum, it's important to show there is no relationship between rank and age in both male and female subjects so we can understand if the effects are likely to be independent of rank.

Second main comment: In study 2, can you provide prior evidence that Barbary macaques attend to differences in conspecific facial expression for printed pictures? Because you detect no effect of picture type on the subjects looking at the photos, it is difficult to interpret whether this is because they don't attend to differences in facial expressions, or don't attend to any pictures of conspecifics. The information in the discussion on macaques attention to conspecifics, babies, friends etc. suggests they can use pictures to identify individuals, but is there any evidence they also interpret facial expression from pictures? If not, what are the implications of this for your results?

Minor comments

In line 31 - can you provide a quick explanation of social selectivity theory for non-experts? For instance, please explain how a limited future time perspective is expected to drive behaviors that enhance well-being under SST.

Line 36 should read, "Both SST and the SAVI models..."

Line 91 - What criteria were used for choosing the 84 study subjects? In other words, why were these animals chosen from the 170-180 animals available? Please include the number of subjects per age category, as you do in line 266 for study 2.

Line 83 - did the "young" category include non-adult animals? In other words, at what age did the young category start? I realize this information is in Table S1, but it would be useful to briefly mention it here as well.

I have a few questions about the modeling decisions. In line 110, can you clarify what the response variable is in these models (e.g. a rate? A count in each sample? A total count per subject?) Doing so will help clarify what's being tested and your choice of the negative binomial error structure.

What was the rationale for not including individual identity as a random effect in models 1 and 2? Likewise, why was a random effect used in model 3, but not model 4? Doubtless there is a good reason, but without explanations, the decisions feel arbitrary.

As in models 1 and 2, please clarify the exact nature of the response variable in model 3.

In your discussion sections, can you briefly discuss whether the captive setting may have played a role in your results, and speculate about whether the results would be the same or different in a natural population?

We thank the reviewers and the editor for the overall positive assessment of our manuscript. Below, we detail how we responded to the comments and suggestions (marked in blue; text passages from the manuscript are given in italics). We hope that we could resolve all issues in a satisfactory manner. We also made some minor changes to the text to improve clarity, where we deemed this necessary. Note that the line numbers refer to the highlighted version of the revised manuscript (PDF). We did not add the new figures to the response letter (per recommendation of the Comm Biol check list), as we only added legends and changed the axis label text (detailed below).

Reviewer #1 (Remarks to the Author):

This is a thoughtful and detailed study examining age-related changes in sociality in macaques. The study appears to be well-powered to support either no or a very tiny effect of age. Overall, the manuscript is well-written and detailed (sometimes too detailed). The major comments relate to the justification and interpretation of this study in the context of two similar previous studies from the same group and the impact of the modeling decisions the authors made, which are detailed below.

Reply: Thank you very much for this positive feedback. The comments helped us to be more explicit in which way the present study goes beyond our previous studies and we feel that the manuscript has gained in quality.

Overall comments:

This work builds upon previous studies from this group showing decreased motivation, but not interest, for social interactions with age (<https://pubmed.ncbi.nlm.nih.gov/28984992/>, <https://pubmed.ncbi.nlm.nih.gov/27345168/>). The novel component here seems to be the focus on the avoidance of agonistic signals, not just interactions, but that is not clearly justified in the introduction.

Reply: The novel angle is the focus on the contingency between signals and responses, while the previous studies mostly investigated physical interactions, including grooming and contact sitting. We expanded the introduction to explain in which the present study builds on and extends previous research. We now write:

“Building on previous studies, specifically the investigation of age-related variation in the occurrence of affiliative and agonistic social interactions in females¹⁶, and age-related variation in male and female social network position¹⁷, we here focussed on the use and responses to communicative signals in both males and females. Communicative signals typically function to initiate or deter subsequent physical interactions such as grooming or physical fighting. Analysing both the usage of signals and the contingency between signal and outcome, that is, whether and in which way the receiver responds to a given signal may provide nuanced insights into age-related changes in the motivation to initiate or avoid physical interactions. To address our research question, we combined an analysis of natural signal exchanges with a field experiment in which we investigated age-related variation in the interest in negative social information.” (line 63 ff)

Overall, it is unclear why the authors expect a decrease in initiation of all interactions, including affiliative, given their previous findings and, importantly, how this fits with the SAVI model they are testing. It might be better to focus just on the negative interactions, which are the crux of their test of the SAVI model. Alternatively, if the authors wish to continue to present changes in affiliative signals then better justification for why both production of and responsiveness to affiliative signals are predicted to decline with age based on SST theory and/or the author's

previous findings is needed. Additionally, it is important that the authors clearly state how this study is different from their previous study(ies) – especially their 2017 paper. In the discussion the authors should clearly explain how these findings either support, extend, or counter their previous findings.

Reply: We do not follow why the reviewer is surprised that we expected a decrease in initiation of all interactions, since we observed indeed fewer social interactions in older animals. We now also explain why we are considering both agonistic and affiliative signals – we think that only the comparison allows us to distinguish whether older animals generally respond less or whether they would specifically respond less to agonistic signals. We now write:

“Although the SAVI model focusses on the avoidance of negative situations, we analysed the production of and responses to both affiliative and agonistic signals, to be able to infer whether the monkeys specifically avoided negative interactions or social information, or whether they generally signalled or responded less.” (line 94 ff)

Reviewer 1: There is a lot of detail about the statistical approach, almost too much, which makes the text a bit more confusing than necessary. For instance, in L139-142 (and elsewhere), there is discussion of a number of different random effects and slopes nested within them– this is a lot of parameters in one model. More importantly, it is unclear why all of these random effects and slopes were included and why none of the slopes themselves were discussed in the results (which is important for interpretation). These random slopes were included in some of the null models, which might explain the lack of a main/fixed effect in the full models. On a related note: what is an “overconfident model”? Why is the signaler age a random slope within the receiver and vice versa? It is even unclear from the code, which is great and available on OSF. Overall these statistical decisions should be justified and clarified – especially when the authors want to report null findings.

Reply: We thank the reviewer for these questions. In keeping with the recommendations and requirements of Communications Biology, we decided to maintain all the statistical details to ensure that the study could be reproduced. We added a sentence about why the inclusion of random slopes is important, namely to prevent from type I errors, that is erroneous “significant” findings. We replaced the term ‘overconfident model’ with ‘overconfident estimates’, the term coined by Forstmeier and Schielzeth 2013, to avoid confusion. Overconfident estimates refer to erroneous significant findings (lines 192 ff).

It is indeed critical to include random slopes, as several studies found highly elevated type I error rates when random slopes are not included (Aarts et al. 2015, Barr et al. 2013 and Schielzeth & Forstmeier 2009). That is, not including random slopes (in the full and the null model) likely leads to spurious findings. We therefore did not change the analysis.

Please also note that all random slopes are included in both the full and null models, and that the crucial test is the full-null model comparison using an anova. In this way, we are testing whether the inclusion of the predictor variable(s) explain the data better than all control variables alone.

Regarding the question of whether the inclusion of “random slopes [...] in some of the null models [...] might explain the lack of a main/fixed effect in the full models”: the reviewer seems to imply that doing so leads to a lack of power and hence potentially erroneously non-significant findings. Given that the main conclusions are based on a full-null model comparison, this point is not of concern.

Including random effects does not lead to a major reduction of power with regard to inference about the contribution of the fixed effects (see Ben Bolker’s GLMM FAQ as of April 6 2022 at

<https://bbolker.github.io/mixedmodels-misc/glmmFAQ.html> where he states that “If a variance component is zero, dropping it from the model will have no effect on any of the estimated quantities [...]” and “Conversely, if one chooses for philosophical grounds to retain these parameters, it won’t change any of the answers.”).

Minor comments:

What is the difference between motivation and interest? And how does that play into this study and their prior work?

Reply: Interest is what we measure, and motivation is the underlying construct that – we assume – drives interest. We tried to be more explicit with regards to the use of these terms. We now write

“Note that ‘social interest’ refers to the behaviour we are measuring (attending to social information), while motivation refers to the underlying construct assumed to drive the expression of social interest.” (line 73 ff)

L108: “Signal usage” is a bit of a vague title, the focus here is on the production of affiliative/agonistic signals, so perhaps something like “Signal production” might be more informative for the reader.

Reply: we would prefer to keep the term “signal usage” because this is the typical term in this context. Signal production could refer to both the structure of the signal and its use; signal usage is more specific and thus clearer, in our view.

L132: Similar to the comment above, a more informative title would be useful here, something like “Probability of responding to affiliative and agonistic signals”. Along the same lines, whatever title is chosen here should align with the title given for that section in the Results (i.e. L208) so that the reader can easily follow the moving pieces of this study.

Reply: thanks for this suggestion; we implemented it (“*Probability of showing any response to group members’ signals*”) and also ensured that the titles are the same in the methods and results section. Sorry for this mistake.

Reviewer 1: L152-156: Did you remove all observations from one individual (or dyad) and then rerun the model to assess stability or did you remove the covariates themselves and assess stability? It is unclear.

Reply: We complemented the sentence, in the hope to make it clearer. We now clearly state that we omitted the individuals one-by-one (line 168)

Reviewer 1: L192: “Varied” in what way? There is no indication that there is a quadratic pattern in how the production of agonistic signals changes with age; the reader has to look to Figure 1 to get this information. On the whole, the figures were clear, but the text was difficult to parse and understand. Better descriptions of the results are needed here and elsewhere in text (e.g. L199). Along the same lines, it would be preferable to present the effect sizes and the p-values for those effect sizes, not just the LRTs for model comparisons. Overall, the results need to focus on the effects, not the model comparisons.

Reply: We agree that this first sentence of the paragraph is not very informative beyond stating the result of the full-null model comparison. Yet, it would not be correct to state that the frequency

increases or decreases with age (for instance), as one would technically need within-subject repeated measurements for such a claim. Likewise, we generally try to avoid causal claims (but do not always manage). To be on the somewhat dry, but safe side, we can only state that older subject produced fewer signals than younger ones – and this statement takes more than half a phrase. Hence, we start with the rather simple statement that the response variable varied with age and sex, and then illustrate the age- and sex-related variation in more detail in the following sentence. Together with the Figure, in our view, these descriptions give a fairly complete picture of what the model revealed.

We now added effect sizes for age and sex at the end of the paragraph. See lines 181 ff for the methods and lines 270f and 268f for the results.

Furthermore, we moved two tables with the results for the analysis of signal usage from the supplemental data to the main manuscript to provide the P-values for the predictor variables. We hope that this satisfies the recommendation for greater clarity in the presentation of the results.

Reviewer 1: L209: The wording here seems misleading. It sounds like there is no overall effect of age (when there is) when what is really meant is that there was no significant interaction between age and signal category. Maybe simplify to what is stated on L210.

Reply: We thank the reviewer for spotting our sloppy wording here. What we meant is receiver age (added). We also clarified that we meant the full null model comparison. We now write “*There was no evidence for the predicted interaction between receiver age and signal category affecting the likelihood to respond*” (line 301).

L218 -227: This section is generally confusing and requires some additional clarification. Why is Fig. 2 referenced here? This figure presents the results from the analysis looking at the probability of responding, not the type of response. Similarly, where did “teeth chatter” come from? In the methods the four types of responses that the authors indicate they are looking at following an agonistic signal are “Give Ground, Make Room, Present, and Lip Smack”. On L224-227 the authors switch back to discussing the “likelihood of responding” which, again, is not what these analyses are about.

Reply: Referencing the Figure 2 here was a mistake – this reference is now deleted. ‘Lip smack’ was also a mistake – it is now corrected. The structure of the text has been cleaned up and the discussion of the likelihood to respond is where it belongs. Thanks for the keen eye!

Figure 2 & 3: Minor comment: put labels on the actual figure panels. E.g., “a) affiliative.” so that the reader doesn’t have to go through the legend to completely understand.

Reply: Done.

Figure 4. To someone who studies rhesus macaques, those look like affiliative images (e.g., “coos”), and not agonistic ones (e.g., open-mouth threats). Especially the female example photo. If both these agonistic photos are indeed open-mouth threats then perhaps a bit more context is warranted for what these behaviors look like in this species?

Reply: Barbary macaques do not produce “coos” and hence there is no risk of mistaking one facial expression from the other. We now say so in the legend to the figure.

L299-301: Could the authors provide some additional descriptive stats for duration of first look and total looking time (e.g. min,max,mean)?

Reply: Because the data are skewed, the mean is actually not appropriate. We therefore provided the median, min, and max values for the initial and total looking time duration (line 470 ff).

We also provided some more information on the relative frequency of different types of responses to provide a better ‘feel’ for the data. We wrote: “*With regard to the control predictors, we found that subjects were more likely to respond with ‘Teeth chatter’, a low-cost submissive signal, in response to agonistic signals by females (42% of N = 860 cases) than to agonistic signals by males (28% of cases). In turn, subjects were more likely to lean or walk away following male agonistic signals (combined ‘give ground’ and ‘make room’: 70% of N = 850 cases) compared to female agonistic signals (55% of cases). A follow-up analysis of the different response types showed that old females were generally more likely to respond to an agonistic signal with ‘Teeth chatter’ compared to younger females (Supplementary Fig. 1d).*”

Reviewer 1: L313: Why is “picture type” included as a random effect nested within signaller? And what is the difference between Picture ID and Signaller ID in this context? Should one of these be Subject ID? It is not clear.

Reply: Each picture was shown two to five times. In order to avoid pseudo-replication at this level, we included Picture ID as an additional random effect. The use of the term ‘Signaller ID’ was indeed misleading and we thank the reviewer for spotting this. What we meant is ‘test subject’ (changed in the manuscript).

L323: Should this read, “thus raising no concerns”?

Reply: indeed. Changed accordingly.

L324: This is repetitive as the same thing is stated on L315-316.

Reply: omitted accordingly.

Reviewer 1: L328: The wording here is again confusing. This is another instance where it would be helpful to clearly state each result and provide the appropriate parameter estimates and p-values. Also, picture type is a random slope within individuals, but then also a fixed effect. Why?

Reply: It is the usual procedure to include a fixed effect and random slopes for the same predictor. The former estimates the ‘average’ (or ‘population level’) effect of a predictor whereas the latter estimates how much the effect of a predictor varies among the levels of a random effect. We first considered writing a tutorial on random slopes in the appendix, but felt it would be misplaced here.

Fig. 5. Y-axis label should be “duration of first look” not “looking time”

Reply: fixed.

L350: Is there not an alternative way of looking at these results? If the idea is to avoid negative/agonistic interactions to avoid high arousal and the associated negative outcomes then surely you actually need to attend to those cues more strongly (or at least not attend to them less)? So might not continued attentiveness to agonistic signals with age be an indication that older individuals are actively trying to avoid negative interactions?

Reply: Of course, one could also test that view. Yet, we were specifically interested in the avoidance of negative arousal, based on the predictions from the SAVI model. Moreover, our own observations suggested that ‘looking away’ is the best strategy to de-escalate a tense situation between an assertive monkey and ourselves, while looking at a threatening monkey quickly leads to escalation. Similarly, we anecdotally had observed that monkeys would stare to the ground in response to another monkeys’ threats, which we considered a de-escalation strategy. Yet, our hunch was not supported by the data.

Reviewer #2 (Remarks to the Author):

This is a clearly written and well-organized paper. The results have the potential to make a useful contribution to social aging in primates and provide important evolutionary context for patterns of human aging.

Reply: Thank you for your appreciation of the paper.

I have two major comments that I would like to see addressed in a revision. Below these, I list several minor comments.

First main comment: In line 115, I was surprised that rank was not included in the model of agonistic signals. The goal of this analysis (model 1) was to test for age signatures in the use of affiliative and agonistic gestures. As the authors explain, rank will be an important predictor of agonism. However, rank is correlated with age in many species. For instance male rank and age are often correlated in polygynous primates such that young adult males have high rank, and rank falls in middle age and older adulthood. This rank pattern could explain the relationship between age and agonistic gestures seen in Fig. 1A. Therefore, it seems important to control for effects of rank in order accurately interpret the age effect on agonistic signals. At a minimum, it’s important to show there is no relationship between rank and age in both male and female subjects so we can understand if the effects are likely to be independent of rank.

Reply: Thanks for raising this point. Theoretically, one would want to include rank as well. However, rank is inferred from the use of agonistic signals, and further aggressive interactions such as lunge, chase, and physical aggression, which are typically preceded or accompanied by agonistic signals. As a result, the analysis would be completely circular. We had noted this point in the manuscript (now line 152) and would therefore prefer to keep the analysis as it is.

Second main comment: In study 2, can you provide prior evidence that Barbary macaques attend to differences in conspecific facial expression for printed pictures? Because you detect no effect of picture type on the subjects looking at the photos, it is difficult to interpret whether this is because they don’t attend to differences in facial expressions, or don’t attend to any pictures of conspecifics. The information in the discussion on macaques attention to conspecifics, babies, friends etc. suggests they can use pictures to identify individuals, but is there any evidence they also interpret facial expression from pictures? If not, what are the implications of this for your results?

Reply: Thank you for raising this important point. From the Barbary macaque data, we cannot conclude whether they do not or cannot distinguish between the different facial expressions. Yet, data from rhesus monkeys suggest that respond differentially to different facial expressions depicted on photos, tentatively suggesting that the Barbary macaques would also be able to do so. We now include a discussion of this point.

We now write *“Yet, it could be possible that the animals were not able to distinguish different facial expressions when shown in a picture. From our data alone, we are unable to decide whether the*

*animals simply did not or were indeed unable to distinguish between the different facial expressions. Yet, rhesus monkeys, *Macaca mulatta*, responded differentially to different facial expressions depicted on photos¹⁴, tentatively suggesting that the Barbary macaques would also be able to do so.” (line 493 ff).*

Minor comments

In line 31 - can you provide a quick explanation of social selectivity theory for non-experts? For instance, please explain how a limited future time perspective is expected to drive behaviors that enhance well-being under SST.

Reply: Thanks for this suggestion. We now added the following sentence “*According to SST, a shrinking time horizon affects the goals that people set for themselves, and favours behaviours that enhance well-being². For instance, in the face of a limited future time, people tend to focus on meaningful social partners and satisfying activities²”.* (line 31 ff).

We hope this example suffices to communicate the core idea of SST.

Line 36 should read, “Both SST and the SAVI models...”

Reply: We do not follow – why plural? We assumed that there was only one SAVI model. And SST refers to “Social selectivity theory”, which does not require an additional ‘model’ at the end, we assumed.

Line 91 - What criteria were used for choosing the 84 study subjects? In other words, why were these animals chosen from the 170-180 animals available? Please include the number of subjects per age category, as you do in line 266 for study 2.

Reply: we included all subjects from two of the study groups in the age range of interest. The third study group spends much time in cliffs and is harder to observe and was thus excluded from the behavioral observations. This information has now been added to the text (line 117 ff).

Line 83 - did the “young” category include non-adult animals? In other words, at what age did the young category start? I realize this information is in Table S1, but it would be useful to briefly mention it here as well.

Reply: information now added to the text. We write: “*all females \geq five years old and all males \geq seven years*”

Reviewer 2: I have a few questions about the modeling decisions. In line 110, can you clarify what the response variable is in these models (e.g. a rate? A count in each sample? A total count per subject?) Doing so will help clarify what’s being tested and your choice of the negative binomial error structure.

Reply: This model used count data. The main reason for choosing a negative binomial error distribution was that in a respective Poisson model the response was clearly overdispersed which leads to an inflated type I error rate. We added one sentence to the manuscript clarifying that the response was indeed a count, but that including focal time as an offset term means that one effectively models signaling rate (i.e., number signals per unit time). We added a statement in brackets a few sentences further up, stating that the response was the total number of signals per signaler in both models.

We now write: “*Although the response was a count in both models, by including focal time as an offset term, we effectively modelled signalling rate (i.e., number signals per unit time).*” (line 166)

Reviewer 2: What was the rationale for not including individual identity as a random effect in models 1 and 2? Likewise, why was a random effect used in model 3, but not model 4? Doubtless there is a good reason, but without explanations, the decisions feel arbitrary.

Reply: as clarified in response to the previous comment and also in the manuscript, the data analysed with model 1 and 2 comprised only one data point per individual. Hence there was no need to deal with pseudo-replication. The data analysed with model 3 and 4, in turn, comprised multiple data points per individual, and hence we needed to control for this. Please note that we included the random effect in model 4, as we clearly state that “*The fitted model was identical to the model of ‘any response’ (model 3; with the exception that it lacked signal category in the fixed as well as random-effects part)*” (line 233).

Reviewer 2: As in models 1 and 2, please clarify the exact nature of the response variable in model 3.

Reply: we added to the very first sentence of this section that the response was just no or yes. (line 186)

In your discussion sections, can you briefly discuss whether the captive setting may have played a role in your results, and speculate about whether the results would be the same or different in a natural population?

Reply: we added the following sentences to the discussion. “*One could raise the question to which degree the monkeys’ signalling and response patterns were influenced by the living conditions. It may well be that the overall interaction frequencies are lower in the wild compared to captivity, when the animals need to spend more time on foraging or group movement. Yet, it seems unlikely that the contingencies between signals and responses would differ substantially, as they appear to be rather fixed across age-classes. Note, however, that an equivalent analysis would not be possible in the wild, as extremely old animals are hardly found under natural conditions.*” (line 533).

Reviewers' comments:

Reviewer #1 (Remarks to the Author):

I continue to believe this paper makes a valuable contribution to the social aging literature. The manuscript is much improved after responses to my comments and those of the other reviewer.

Reviewer #2 (Remarks to the Author):

I thank the authors for their thorough response to the reviewer comments. The changes the authors have made have better positioned their paper within the context of their previous findings and have provided a clearer understanding of their modeling decisions. I remain convinced that the author's methodology is robust, their analyses are sound, and that this manuscript will be a nice addition to the social aging literature. However, there are still several points which would benefit from additional clarification. I list these below.

L72-76 – I appreciate the authors offering additional context as to why they are looking at decreases in agonistic AND affiliative interactions (e.g. L82-85). I agree with the authors that it makes good sense to look at both. However, the way in which this is written on Lines 72-76 is still misleading. It sounds as though the predictions for older individuals being less likely to initiate or respond to affiliative signals follows directly from the SAVI model, which (to the best of my understanding) it does not. As far as I can tell from the introduction, the SAVI model predicts a decrease in attention to or production of agonistic signals but does not make such predictions for affiliative signals. While the predictions for declines in affiliative signals might follow logically from the authors' previous findings, the underlying rationale for these different predictions (i.e. affiliative declines based on previous findings vs. agonistic declines based on SAVI model) need to be made more clear. (Also, given the fact that older macaques maintain interest in the social environment (Almeling et al. 2016) it is not entirely clear to me why the authors would necessarily predict a decline in responsiveness to affiliative signals. Is this not a measure of social interest – i.e. a measure of "attending to social information"? Or do the authors see responding to affiliative signals as a measure of willingness to 'engage in' social interaction?)

L81 – In line with the comment above, should the prediction here be that older monkeys would spend less time looking at BOTH agonistic and neutral pictures, but that the effect would be more pronounced in the agonistic stimuli? This is what lines 72-76 seem to imply, that the authors expect a decrease in attentiveness to all social stimuli with age, but that the decrease is more pronounced for agonistic stimuli. I can see why the authors might make an alternative prediction (i.e. that older monkeys would spend less time looking at the agonistic photos only), given interest in social stimuli has been shown not to change in the Barbary macaques with age. I think that some clarification as to what these different studies are aiming to measure might be helpful – e.g. does Study 1 reflect willingness to engage in agonistic/affiliative interactions and Study 2 reflect attentiveness toward affiliative/agonistic social information? To me, both responsiveness to behavioural signals (study 1) and responsiveness to visual stimuli (study 2) reflect social interest and so I would expect the predictions to be the same.

L124 – I apologize for being picky, but I still found this section somewhat difficult to follow in and I just want to encourage the authors to clarify the structure a bit to help their readers better follow their findings. When discussing 'signal usage' the authors are referring specifically to the effect of age on how often individuals produce signals, is that correct? Then in the section termed "Probability of showing any response to group members' signals" (L155) the authors have switched to talking about how age affects the responsiveness to signals, correct? If so, then perhaps an additional sub-header is warranted? Something like "Signal responsiveness" which would be a sub-header in the "Statistics and Reproducibility" section and "Probability of showing any response to group members' signals" as well as "Type of response" would be sub-sub-headers below that. This would help to differentiate the analyses that focus on "production of" vs. "responsiveness to" signals.

L164-167 – I appreciate the authors explaining more clearly why they have included so many

random slope terms in their models. I disagree a bit with this approach in that I think random slopes should only be included in the model if there is a sound biological rationale for doing so. However, I agree with the authors in that the inclusion of these terms, given how their data are analyzed, is unlikely to affect the results. However, in my opinion it would still be useful to provide a clearer explanation of what each of these random slope terms actually represents. E.g. the inclusion of signaller age as a random slope over receiver ID estimates how much among-individual variation there is in how a receiver respond the age of the signaller. In my opinion, understanding what these random slope terms actually represent biologically is important for understanding why their inclusion in the model might be important and what variation that term is accounting for.

L183-190 – I found this section difficult to follow. Is the upshot of this section that model 3 only included females and not males? If so, this should be stated earlier and more clearly. If not, then there needs to be more consistency in the descriptive stats reported for females and males. E.g. Why is the number of signallers and receivers and dyads not given for males? Why is the number of responses for males split between affiliative and agonistic events while the combined numbers are given for females?

L199-200 – It is not clear what kind of model this was, only that it was fit in the brms package. Maybe calling this a “multinomial” model is sufficient but some additional information would be helpful here as I’m not familiar with models that take categorial response variables.

L216 – I appreciate the authors incorporating their results tables in text here as that was very helpful for interpreting their findings. However, why is information for only one LRT reported in text? Wasn’t there a separate test done for each fixed effect so shouldn’t there be two LRTs reported here?

L234-237 – I did not follow this sentence, could the authors reword/clarify?

L239/Table 2 – Shouldn’t the coefficient estimate for age be negative here?

L244 – I found it difficult to jump into these results. Perhaps an introductory sentence here to remind the reader what these analyses were looking at would be helpful.

L244 – Could the authors put “signal category (i.e. affiliative/agonistic)” here, just to remind the reader what “signal category” is referring to? I was confusing this with the “type of response” that is discussed in the next section.

L248 – Get rid of “however” as it doesn’t seem appropriate and put a semicolon after “signaller age”.

L251 – Wording. Consider something like: “The likelihood of responding did not vary either with signaller sex or signal category”.

L269 – What “control predictors”? Be specific here. For example, “We found that the type of response was related to the sex of the signaler...”

L274- It is not clear what this “follow-up analysis” was or how it differed from the original analysis. I couldn’t find any mention of it in the methods.

L351-352 – Same comment as above regarding describing what the random slopes mean.

L366-367 – Wording. Consider something like: “Neither age, picture type, or the interaction between age and picture type was clearly related to variation in initial looking time.”

L419-421 – I’m a bit confused by this statement that overall motivation to engage in social interactions declines with age. Maybe this is because I remain bit confused by the use of the terms “social motivation” vs. “social interest”. As I understand it, the authors have shown that (in line with their previous findings) interest in social information remains consistent into old age. That is,

they did not find a decrease in the probability of responding to affiliative or agonistic signals with age (Fig. 2) nor did they find a decrease in old individuals' responsiveness to agonistic or neutral visual stimuli (Fig. 5). Both findings seem to suggest that old individuals remain interested in their social world (see also L387). If social motivation is, as the authors define it, "the underlying construct assumed to drive the expression of social interest", then the fact that social interest is maintained into later life would suggest that social motivation is also maintained into old age. The authors do find a decline in the usage of affiliative and agonistic social signals in older individuals, perhaps suggesting a decline in willingness to engage in the social world with age. Maybe this is what they are referring to when they say "social motivation" here? But if so, there seems to be some inconsistency in how they are using the term.

Reviewers' comments:

Reviewer #1 (Remarks to the Author):

I continue to believe this paper makes a valuable contribution to the social aging literature. The manuscript is much improved after responses to my comments and those of the other reviewer.

Response: *We thank the reviewer for the positive assessment.*

Reviewer #2 (Remarks to the Author):

I thank the authors for their thorough response to the reviewer comments. The changes the authors have made have better positioned their paper within the context of their previous findings and have provided a clearer understanding of their modeling decisions. I remain convinced that the author's methodology is robust, their analyses are sound, and that this manuscript will be a nice addition to the social aging literature. However, there are still several points which would benefit from additional clarification. I list these below.

Response: *We thank the reviewer for this overall positive assessment. In our revision, we tried to address all of the suggestions and now hope that the reviewer will be satisfied with the revised version.*

L72-76 – I appreciate the authors offering additional context as to why they are looking at decreases in agonistic AND affiliative interactions (e.g. L82-85). I agree with the authors that it makes good sense to look at both. However, the way in which this is written on Lines 72-76 is still misleading. It sounds as though the predictions for older individuals being less likely to initiate or respond to affiliative signals follows directly from the SAVI model, which (to the best of my understanding) it does not. As far as I can tell from the introduction, the SAVI model predicts a decrease in attention to or production of agonistic signals but does not make such predictions for affiliative signals. While the predictions for declines in affiliative signals might follow logically from the authors' previous findings, the underlying rationale for these different predictions (i.e. affiliative declines based on previous findings vs. agonistic declines based on SAVI model) need to be made more clear.

Response: *We have changed the wording and now clearly distinguish between the analysis that is driven by predictions of the SAVI model, and the ones that we added to control for overall decline in motivation to respond to signals.*

We now write: "We expected that older Barbary macaques would be less likely to initiate social interactions using communicative signals than younger monkeys. Following the SAVI model, we predicted that this effect would be more pronounced for agonistic signals. We further predicted that older monkeys would be less responsive to other monkeys' signals. Again, following the SAVI model, we expected that the effect would be more pronounced for agonistic signals. Specifically, we expected that older monkeys would be more likely to adopt avoidance or de-escalation strategies, such as ignoring others' signals or leaning or walking away (study 1). Considering the production of and responses to affiliative signals was deemed crucial, because it allowed us to distinguish between general age-related changes in signalling vs. the specific avoidance of negative interactions." (L 77)

(Also, given the fact that older macaques maintain interest in the social environment (Almeling et al. 2016) it is not entirely clear to me why the authors would necessarily predict a decline in responsiveness to affiliative signals. Is this not a measure of social interest – i.e.

a measure of “attending to social information”? Or do the authors see responding to affiliative signals as a measure of willingness to ‘engage in’ social interaction?)

Response: *We now write in the introduction: “Note that we distinguish between the motivation to acquire social information (‘social interest’) and the motivation to engage in physical social interactions. We assume that social interest is a precondition for the willingness to engage in physical social interaction, but is not necessarily tied to it. In other words, the animals may show social interest but may not be motivated to come into body contact, or groom, or fight with others. We assume that the exchange of social signals plays a decisive role in increasing or decreasing the likelihood of physical interactions.” (L 69 ff)*

In the discussion, we added: “Overall, our results as well as those by Rosati and colleagues 15 suggest that the general motivation to engage in social interactions declines with age in both humans and different nonhuman primate species, although the motivation to acquire social information does not vary with age (this and other studies on nonhuman primates). With regard to the management of social interactions, the available evidence suggests that only humans are able to employ more sophisticated self-regulation strategies in old age” (L 424 ff)

L81 – In line with the comment above, should the prediction here be that older monkeys would spend less time looking at BOTH agonistic and neutral pictures, but that the effect would be more pronounced in the agonistic stimuli? This is what lines 72-76 seem to imply, that the authors expect a decrease in attentiveness to all social stimuli with age, but that the decrease is more pronounced for agonistic stimuli. I can see why the authors might make an alternative prediction (i.e. that older monkeys would spend less time looking at the agonistic photos only), given interest in social stimuli has been shown not to change in the Barbary macaques with age. I think that some clarification as to what these different studies are aiming to measure might be helpful – e.g. does Study 1 reflect willingness to engage in agonistic/affiliative interactions and Study 2 reflect attentiveness toward affiliative/agonistic social information? To me, both responsiveness to behavioural signals (study 1) and responsiveness to visual stimuli (study 2) reflect social interest and so I would expect the predictions to be the same.

Response: *As noted above, we assume that signaling before actual physical interaction and attending to social information in the environment are different categories for us. This distinction goes back to the Almeling et al. 2016 paper, where we clearly distinguished between social interaction and attending to screams (in playbacks) or pictures, which we classified as “social interest”. Signaling is more clearly tied to physical proximity with another subject and thus, the potential costs differ substantially between the two categories (they are potentially higher following a signal exchange compared to attending to somebody’s screams behind bushes).*

L124 – I apologize for being picky, but I still found this section somewhat difficult to follow in and I just want to encourage the authors to clarify the structure a bit to help their readers better follow their findings. When discussing ‘signal usage’ the authors are referring specifically to the effect of age on how often individuals produce signals, is that correct?

Response: *Yes, we are referring to signal production. We had tried to explain why we prefer the term usage over production, but since our choice of terminology was not convincing, we have now changed the wording from usage to production. The header now reads: “Signal production in relation to age”. We changed the rest of the text accordingly.*

Then in the section termed “Probability of showing any response to group members’ signals” (L155) the authors have switched to talking about how age affects the responsiveness to signals, correct?

Response: Yes

If so, then perhaps an additional sub-header is warranted? Something like “Signal responsiveness” which would be a sub-header in the “Statistics and Reproducibility” section and “Probability of showing any response to group members’ signals” as well as “Type of response” would be sub-sub-headers below that. This would help to differentiate the analyses that focus on “production of” vs. “responsiveness to” signals.

Response: *Thank you for this recommendation. We added the recommended sub-headings and hope it will now be easier to navigate the text.*

L164-167 – I appreciate the authors explaining more clearly why they have included so many random slope terms in their models. I disagree a bit with this approach in that I think random slopes should only be included in the model if there is a sound biological rationale for doing so. However, I agree with the authors in that the inclusion of these terms, given how their data are analyzed, is unlikely to affect the results.

However, in my opinion it would still be useful to provide a clearer explanation of what each of these random slope terms actually represents. E.g. the inclusion of signaller age as a random slope over receiver ID estimates how much among-individual variation there is in how a receiver respond the age of the signaller. In my opinion, understanding what these random slope terms actually represent biologically is important for understanding why their inclusion in the model might be important and what variation that term is accounting for.

Response: *We are now spelling out the biological significance for two of these terms and also added the biological implication of not including random slopes using the same example in the main manuscript. In this way, we hope to explain also why we deem it not only theoretically recommendable but indeed biologically important to include these random slopes.*

We now write: “In terms of their biological meaning, these random slopes estimate the extent to which the effect of a predictor varies between signallers or receivers. For instance, the random slope of receiver age within signaller takes into consideration the possibility that the responsiveness to a signal will vary in an age-related fashion with the identity of the signaller (e.g., older subjects will be more or less likely to respond to subject A vs. subject B). Similarly, the random slope of signaller age within receiver estimates how much a potential effect of signaller age on responsiveness varies between receivers. Importantly, not including the respective random slope means to make the strong (unlikely) assumption that, for instance, the effect of signaller age on responsiveness is exactly the same for all receivers.” (L 175)

L183-190 – I found this section difficult to follow. Is the upshot of this section that model 3 only included females and not males? If so, this should be stated earlier and more clearly.

Response: *We clearly spelled it out that for this analysis, only females were used. We now moved this information up in the text, and explained why we confined this analysis to females (small sample size for males). (L 163)*

If not, then there needs to be more consistency in the descriptive stats reported for females and males. E.g. Why is the number of signallers and receivers and dyads not given for males? Why is the number of responses for males split between affiliative and agonistic events while the combined numbers are given for females?

Response: *As stated above, this model was only run for females.*

L199-200 – It is not clear what kind of model this was, only that it was fit in the brms

package. Maybe calling this a “multinomial” model is sufficient but some additional information would be helpful here as I’m not familiar with models that take categorical response variables.

Response: *We added a statement about the relation between logistic and multinomial models, also added a reference which considers multinomial-models in detail and hope that this is sufficient. We wrote: “However, the response in this model was a categorical variable with four states. Hence, we fitted a multinomial model, which can be conceived as a generalization of the logistic model suited for a response comprising more than two states.” (L 215).*

L216 – I appreciate the authors incorporating their results tables in text here as that was very helpful for interpreting their findings. However, why is information for only one LRT reported in text? Wasn’t there a separate test done for each fixed effect so shouldn’t there be two LRTs reported here?

Response: *The Likelihood Ratio Test refers to the full-null model comparison. We are sorry that this did not become clear before. We have now specified this information in the text.*

L234-237 – I did not follow this sentence, could the authors reword/clarify?

Response: *We rewrote this sentence. It now reads: “Note that the estimate for males was negative (i.e., males should be less likely to produce affiliative signals than females). Yet, an inspection of the data shows that males were indeed more likely to produce affiliative signals. This paradox can be explained by the fact that males were on average higher ranking than females, such that the positive effect of rank explained also the variation in signal likelihood between males and females, with higher ranking animals being more likely to produce affiliative signals. In conjunction, males thus produced fewer signals than expected for their rank position.” (L 250 ff)*

We hope that the issue is now clarified.

L239/Table 2 – Shouldn’t the coefficient estimate for age be negative here?

Response: *Indeed, that was a typo! Thanks for spotting this error (the confidence intervals had the correct sign).*

L244 – I found it difficult to jump into these results. Perhaps an introductory sentence here to remind the reader what these analyses were looking at would be helpful.

Response: *We thank the reviewer for this suggestion. We now added the following introductory sentence: “In the following analysis, we investigated whether older animals were particularly likely to ignore agonistic signals.” (L260 f)*

L244 – Could the authors put “signal category (i.e. affiliative/agonistic)” here, just to remind the reader what “signal category” is referring to? I was confusing this with the “type of response” that is discussed in the next section.

Response: *We changed the text accordingly.*

L248 – Get rid of “however” as it doesn’t seem appropriate and put a semicolon after “signaller age”.

Response: *We changed the text accordingly.*

L251 – Wording. Consider something like: “The likelihood of responding did not vary either with signaller sex or signal category”.

Response: *According to the Cambridge Dictionary, “neither nor” is used for negative lists (as in “signaller sex did not affect the likelihood to respond, and signal category also did not have an effect”), not “either or” – perhaps the Editor can have the last word here? For now, we did not follow the suggestion.*

L269 – What “control predictors”? Be specific here. For example, “We found that the type of response was related to the sex of the signaller...”

Response: *We now write: “With regard to the control fixed effects ‘signaller age’ and ‘signaller sex’...”*

L274- It is not clear what this “follow-up analysis” was or how it differed from the original analysis. I couldn’t find any mention of it in the methods.

Response: *We thank the reviewer for pointing this issue out. Indeed, this was not a follow-up analysis but a further inspection of the outcome of the multinomial model. We now write “In the multinomial model, we also tested whether the different response types varied differentially with age. Old females were more likely to respond to an agonistic signal with ‘Teeth chatter’ compared to younger females, while there was no age effect for the other signal categories“. (L 279)*

L351-352 – Same comment as above regarding describing what the random slopes mean.

Response: *We hope that the explanation mentioned above is now sufficient to allow the reader to figure out what the random slopes mean.*

L366-367 – Wording. Consider something like: “Neither age, picture type, or the interaction between age and picture type was clearly related to variation in initial looking time.”

Response: *The suggested version mixes “neither” with “or”. We checked with the Cambridge Dictionary and “neither” – “nor” would be the correct version. We thus did not change the sentence as suggested.*

L419-421 – I’m a bit confused by this statement that overall motivation to engage in social interactions declines with age. Maybe this is because I remain bit confused by the use of the terms “social motivation” vs. “social interest”. As I understand it, the authors have shown that (in line with their previous findings) interest in social information remains consistent into old age. That is, they did not find a decrease in the probability of responding to affiliative or agonistic signals with age (Fig. 2) nor did they find a decrease in old individuals’ responsiveness to agonistic or neutral visual stimuli (Fig. 5). Both findings seem to suggest that old individuals remain interested in their social world (see also L387). If social motivation is, as the authors define it, “the underlying construct assumed to drive the expression of social interest”, then the fact that social interest is maintained into later life would suggest that social motivation is also maintained into old age. The authors do find a decline in the usage of affiliative and agonistic social signals in older individuals, perhaps suggesting a decline in willingness to engage in the social world with age. Maybe this is what they are referring to when they say “social motivation” here? But if so, there seems to be some inconsistency in how they are using the term.

Response: *Please see above for our expanded section on the distinction between social interest and the motivation to engage in physical interaction.*